# Ventilation of the Northern Baltic Sea

Thomas Neumann[1], Herbert Siegel[1], Matthias Moros[1], Monika Gerth[1], Madline Kniebusch[1], and Daniel Neumann[1]

[1]Leibniz Institute for Baltic Sea Research Warnemünde, Rostock 18119 Warnemünde Seestr. 15, Germany

**Correspondence:** Thomas Neumann (thomas.neumann@io-warnemuende.de)

**Abstract.** The Baltic Sea is a semi-enclosed, brackish water sea in northern Europe. The deep basins of the central Baltic Sea regularly show hypoxic conditions. In contrast, the northern parts of the Baltic Sea, the Bothnian Sea and Bay, are well oxygenated. Lateral inflows or a ventilation due to convection are possible mechanisms for high oxygen concentrations in the deep water of the northern Baltic Sea.

In March 2017, CTD profiles and bottle samples, ice core samples, and brine were collected in the Bothnian Bay. In addition to hydrographic standard parameters, light absorption have been measured in all samples. A complementary, numerical model simulation provides quantitative estimates of the spread of newly formed bottom water. The model uses passive and age tracers to identify and trace different water masses.

Observations indicate a recent ventilation of the deep bottom water at one of the observed stations. The analysis of observations and model simulation show that the Bothnian Bay is ventilated by dense water formed due to mixing of Bothnian Sea and Bothnian Bay surface water initializing lateral inflows. The observations show the beginning of the inflow and the model simulation demonstrates the further northwards spreading of bottom water. These events occur during winter time when the water temperature is low. Brine rejected during ice formation barely contributes to dense bottom water.

## 1  Introduction

The Baltic Sea is a semi-enclosed marginal sea in northern Europe with a positive fresh water budget (e.g., Lass and Matthäus, 2008), and as a result, the surface salinity ranges from $14\,\mathrm{g\,kg^{-1}}$ (south) to $2.5\,\mathrm{g\,kg^{-1}}$ (north). Owing to its hydrographic conditions, an estuarine-like circulation is established and a strong and permanent vertical density stratification occurs. These hydrographic conditions set the prerequisites for vulnerability to hypoxia and anoxia below the pycnocline. In general, a ventilation is only possible by lateral intrusions of oxygenated water of a sufficiently high density which allows this water to enter depths below the pycnocline. Indeed, the modern Baltic Sea shows wide areas of hypoxia in its central part. However, the northern part of the Baltic Sea, the Bothnian Sea and Bothnian Bay, are characterized by well oxygenated bottom water although for the last decades a decreasing oxygen concentration was reported (Raateoja, 2013).

The common prerequisite for deep water renewal due to inflows (cascading) is a sufficient lateral density gradient and slope of the ocean floor. A comprehensive summary of possible mechanisms producing density gradients and theoretical aspects for the initiation of cascades is given in Shapiro et al. (2003). In case of the northern Baltic Sea, salinity difference is the main

driver for lateral density gradients. Small temperature differences close to the temperature of maximum density during winter do not have a significant effect on density compared to the effect of observed salinity gradients.

The bathymetric structure of the Baltic Sea is characterized by connected basins separated by shallow sills (Fig. 1). Cascades of descending water may be initiated if the density of water over the sills exceeds the density of the adjacent basin sufficiently. The depth of final interleaving depends on the lateral density difference and the entrainment on its way by the dense water plume. Stigebrandt (1987) estimates the entrainment into dense bottom currents of the Baltic Sea as a function of the slope (eq. 3.12 in (Stigebrandt, 1987)). For the central Baltic Sea, Major Baltic Inflows (e.g., Mohrholz, 2018) are the dominating process for bottom water ventilation. The origin of the new water mass is the North Sea; it leaves the surface in the western Baltic Sea and spreads northwards as a dense bottom plume.

Marmefelt and Omstedt (1993) investigate processes for deep water renewal in the northern Baltic Sea and conclude that inflows of dense water are the main process rather than vertical convection. In contrast to Major Baltic Inflows in the central Baltic Sea, those inflows into the Bothnian Sea and Bothnian Bay originate from surface water of the Aland Sea and Northern Quark (Fig. 1), respectively. Therefore, this inflowing water is saturated with oxygen and these events may occur more often than Major Baltic Inflows. Other possible processes, thermal and haline convection, have been assessed as highly unlikely. However, Marmefelt and Omstedt (1993) also notice that more observations from the Bothnian Bay are needed during winter time.

Geological studies from the Baltic Sea (Moros et al., 2020) indicate that deep water formation cause widespread re-suspension and transport of sediment during colder climatic periods. Sediment contourite drifts are prominent in depths greater than 200m in the Gulf of Bothnia and the northern Baltic Proper and therefore, cold conditions appear to favor dense water production. In fact, one cannot compare today's climate with conditions during the Little Ice Age (e.g., Kabel et al., 2012) when widespread sea ice dominated the Baltic Sea in winter. Today, the Baltic Sea shows conditions where sea ice appears regularly and surface water is cooled down to freezing temperature in the Bothnian Bay only. However, a mechanistic description of the detected sediment redistribution is still lacking.

In addition to dense water formation due to saline water from adjacent seas as described by Marmefelt and Omstedt (1993), dense water also can be formed due to brine from sea ice formation. It is an usual phenomenon in Arctic and Subarctic regions. Dense and saline water masses are generated at hot spots of sea ice production, the polynyas. But also melting sea ice can produce considerable salt fluxes into the ocean. Peterson (2018) observed a substantial salt loss in Arctic sea ice due to warming. The explanation is an increased permeability which allows gravity drainage. The general idea of new water mass formation due to brine is that dense water is generated at the shelf and then a gravity driven flow can reach deeper regions of the ocean (Aagaard et al., 1981; Ivanov et al., 2004; Skogseth et al., 2008). However, owing to the rough weather conditions, experimental studies are sparse.

Sea ice in the northern Baltic Sea shows a considerably lower bulk salinity compared to sea ice in the Arctic ocean due to the low sea water salinity. Usually, values less than $1\,\mathrm{g\,kg^{-1}}$ have been observed (e.g., Meiners et al., 2002; Granskog et al., 2005). In addition, Baltic Sea ice can consist of up to 35% metamorphic snow (Granskog et al., 2006) and therefore, the brine volume in Baltic Sea ice is smaller than in Arctic sea ice. However, brine salinity is comparable (Assur, 1958).

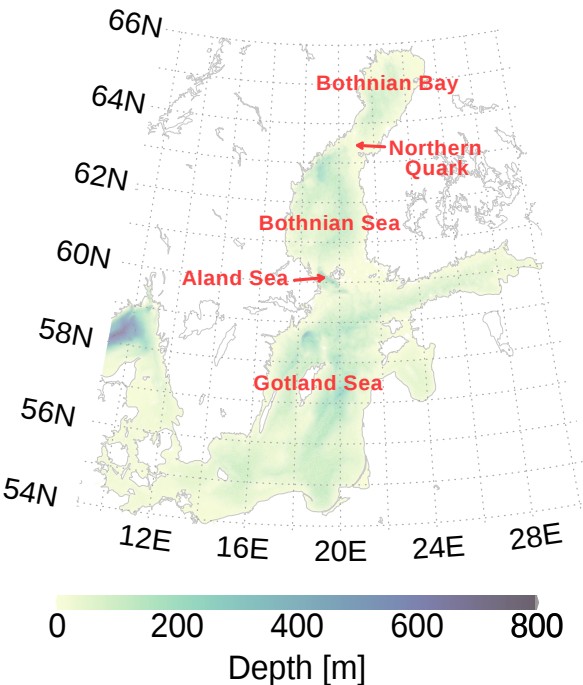

**Figure 1.** Bathymetry of the Baltic Sea showing names of different geographic regions we will use in the text. The map was created using the software package GrADS 2.1.1.b0 (http://cola.gmu.edu/grads/), using published bathymetry data (Seifert et al., 2008).

In this study, we present data from a two days long expedition with the RV *Maria S. Merian* in the sea-ice-covered Bothnian Bay, the most northern basin of the Baltic Sea (Fig. 1), in March 2017. Samples from sea ice, brine, and water column profiles and bottle samples have been taken and analyzed. Furthermore, we performed numerical model studies to reproduce the campaign and to provide additional data for analysis. The aim of this study is to identify relevant processes ventilating the deep water of the Bothnian Bay and to quantify the importance of brine for dense water formation.

## 2   Methods

The Baltic Sea ice season 2016-2017 was mild with a maximum ice extent of $80,000 \mathrm{km}^2$ reached at February 12th (Baltic Icebreaking Management, 2017). Sea ice samples and water samples in the northern Baltic Sea, the Bothnian Bay and the Bothnian Sea, were taken during the expedition MSM62 with RV *Maria S. Merian* between 12th and 13th March 2017. In this time, the Bothnian Bay was well covered by sea ice (Fig. 2). For sea ice sampling, the RV sailed into the ice cover until it was surrounded by pack ice. In a distance of about 50m from the RV, ice core samples were taken. Nearby of the ice core stations,

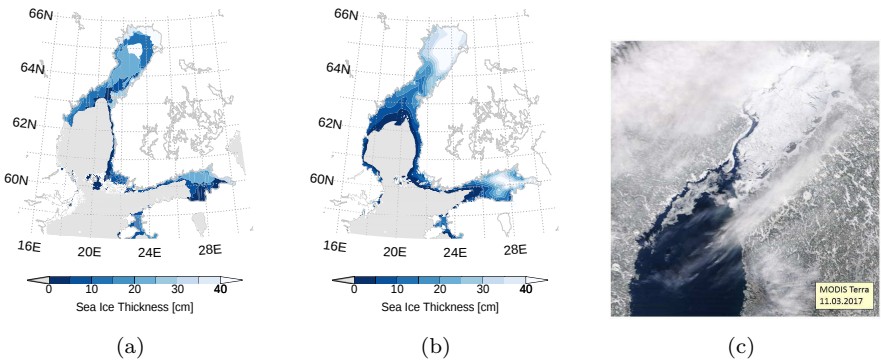

(a)                                    (b)                                    (c)

**Figure 2.** Sea ice thickness (a) at March 12th 2017 in the Bothnian Sea and the Bothnian Bay. Data are based on the Finnish Meteorological Institutes ice charts and freely distributed by the Copernicus Marine Environment Monitoring Service (marine.copernicus.eu). (b) is the same as (a) but from our model simulation. Sea ice coverage (c), quasi true color image derived from MODIS at March 11th 2017. The map in (a and b) was created using the software package GrADS 2.1.1.b0 (http://cola.gmu.edu/grads/)

shipborne CTD measurements were performed. For this purpose, the RV steamed a short distance to be freed from surrounding sea ice and the CTD probe could be lowered undisturbed. Names and locations of stations are shown in Fig.3.

## 2.1   Ice core sampling

With the aid of an ice corer, three to four sea ice cores were taken at each station. The length of the cores varied from 0.3
to 0.6 m. Obvious snow and loose parts were removed from the surface. In addition, holes approximately half the depth of the ice thickness were drilled; the ice core was removed and brine draining from the ice accumulated in the hole. After about 30 minutes, we took the brine sample from the borehole with a pipette. Brine could be collected at two stations (stat. 8 and 10, Fig. 3).

Onboard, the ice cores were cut into two or three segments depending on the total length and structure of the ice core. Snow
was removed from the ice fragments and then the ice was melted for further analysis. After salinity estimations, the melt water was filtered and samples were prepared for analysis of yellow substances. Salinity of the ice cores and brine was measured with a hand-held CTD (see section 2.2). Owing to the size of the sensor pack, a small sample volume of about 100ml was sufficient. In addition, selected salinity samples (Tab. 1) were measured with a Guildline's Autosal 8400B Laboratory Salinometer. The accuracy of the Autosal is better than 0.002 Equivalent Practical Salinity Units (PSU)

## 2.2   Water column sampling

A CTD-system "SBE 911plus" (SEABIRD-ELECTRONICS) was used to measure pressure, temperature, conductivity, oxygen, fluorescence chlorophyll, back-scattering turbidity, and CDOM (colored dissolved organic matter, fluorescence method 370/460nm ex/em). For temperature, conductivity, and oxygen, two sensors were installed to ensure a high standard of data

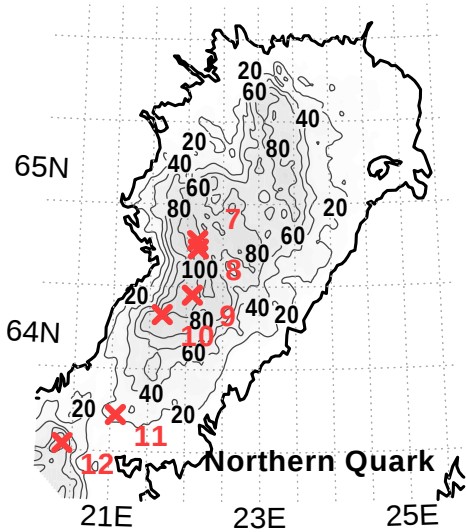

**Figure 3.** Locations of stations (red crosses) in the Bothnian Bay (stations 7 to 10), the Northern Quark (station 11), and the Bothnian Sea (station 12). Stations 7, 11, and 12 were CTD only stations. At stations 8, 9, and 10 sea ice and brine samples were taken. Nearby stations 9 and 10 also a CTD cast was performed. The map was created using the software package GrADS 2.1.1.b0 (http://cola.gmu.edu/grads/), using published bathymetry data (Seifert et al., 2008).

quality. A benthic altimeter delivered the bottom distance. Additionally, the CTD-probe was equipped with an SBE 32 water sampler with 13 free flow bottles of $5\mathrm{dm}^3$ volume. The CTD was always put for at least 3 minutes at 10m depth into the water before the cast started in order to remove air bubbles from the pumping system and water samplers. The CTD was lowered at 0.3 to $0.5\mathrm{m\,s}^{-1}$.

We re-calibrated each oxygen profile with the assumption that the upper 5m are well mixed and saturated with oxygen, i.e. a small offset has been applied to all oxygen profiles. This assumption was made on the basis of temperature and salinity profiles (Fig. 7) which are homogeneous in the upper layer.

    After sea ice coring, a hand-held CTD was deployed through the borehole to measure the water column directly below the sea ice. For these measurements, a CTD48M by Sea & Sun Technology was used. The CTD48M is a very small multi-parameter

probe for precise online measurements. Data is stored internally and can be downloaded after the mission. Owing to the low weight of 1.2kg and the small housing diameter of 48mm, the probe is well suited for deployments in a borehole.

    The probe was equipped with a pressure, temperature, and conductivity sensor. The pressure transducer is a piezo-resistive full bridge. For temperature measurements, a platinum resistor with nominal 100 Ohm resistance at 0°C (PT100) is used. Conductivity is measured by a cell which consists of a quartz glass cylinder with 7 platinum coated ring electrodes.

All shown quantities, like conservative temperature, absolute salinity, freezing temperature, density etc., have been computed following the TOES–10 manual (ICO et al., 2010) from measured in situ data.

## 2.3 Seawater optics

For the determination of the spectral absorption of dissolved organic substances (CDOM, yellow substances), seawater was filtered under low vacuum through Whatman GF/F glass fiber filters (pore size approximately 0.7μm). The filtered water was measured in a 10cm cuvette using a dual-beam Perkin Elmer Lambda 2 instrument in the wavelength range between 300 and 750nm with increments of 1nm. Milli-Q water was the reference. Comparisons between Whatman GF/F and membrane filters with a pore size of 0.2μm did not deliver significant differences for the area of investigation.

The spectral absorption coefficients ay($\lambda$) were calculated according to Kirk (1994):

$$ay(\lambda) = 2.3026(A(\lambda) - A(720\text{nm})/l,$$

where A($\lambda$) is the spectrophotometer absorbance at wavelength $\lambda$, l is the optical path length, and A(720nm) is the baseline correction. The spectral dependence of CDOM absorption is characterized by an exponential increase to shorter wavelengths with a maximum in the UV spectral range and can be described according to Jerlov (1976), and Kirk (1994) by the following equation:

$$ay(\lambda) = ay(\lambda_0) \exp(-s(\lambda - \lambda_0)),$$

where ay($\lambda$) is the absorption coefficient at the wavelength $\lambda$, $\lambda_0$ is the reference wavelength and s the spectral slope for the exponential dependence. The absorption coefficient at 440nm is used for comparisons.

## 2.4 Model simulations

For the analysis of different water masses, we use a 3-dimensional, numerical model of the Baltic Sea. The model code is MOM5.1 (Griffies, 2004) adapted for the Baltic Sea. The horizontal resolution is one nautical mile. Vertically, the model is resolved into 152 layers with a layer thickness of 0.5m at the surface and gradually increasing with depth up to 2m.

The circulation model MOM5.1 is coupled with a sea ice model (Winton, 2000) which accounts for ice formation and drift. Brine rejection is simulated by a sea surface salt flux. With a prescribed bulk ice salinity, the surplus salt is rejected immediately during freezing. Owing to the numerics of ocean models (grid spacing in the order of kilometers and the hydro-static approximation), the models cannot resolve microscopic processes like brine release. Therefore, we implemented a parametrization for the subgrid-scale convection of brine proposed by Nguyen et al. (2009). The parametrization distributes rejected brine within the mixed layer vertically simulating convective salt plumes. We used the calibration of Nguyen et al. (2009) for this study.

For earmarking water masses, we used the capability of the MOM5.1 model to define passive tracers which are set to unity at the surface at each model time step. Two passive tracers, each for the Bothnian Bay and the Bothnian Sea, are set up. Figure 4 shows the horizontal and vertical extent of the passive tracers. Owing to vertical mixing, the passive tracer is evenly distributed over the mixed layer after a few days of simulation. In addition, we have defined an age tracer for each passive tracer. The age of the corresponding tracer is set to zero at the surface at each model time step and otherwise increases with each time step. Furthermore, we defined a brine tracer for each passive tracer region. The brine tracer represent the salinity rejected from sea ice during freezing. That is, the brine tracer is the salinity fraction due to brine.

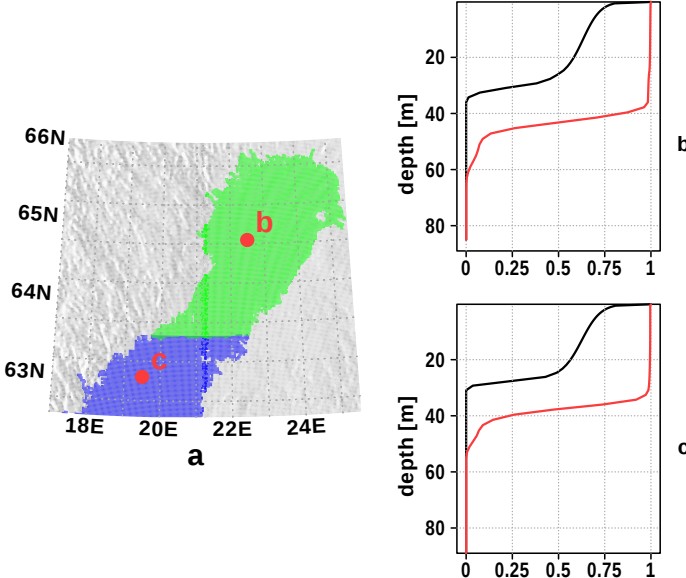

**Figure 4.** Initializing of the surface tracer in the northern Baltic Sea: Horizontal extent a) of Bothnian Sea surface tracer (blue) and Bothnian Bay surface tracer (green), and vertical extent at b) and c). Shown is the vertical profile after 1 day (black) and after 10 days (red).

The model has been forced by meteorological data from the coastDat-2 data set (Geyer and Rockel, 2013). We run the model from 1948–2018 and the passive tracer have been activated in January 2017. In many studies, the model has been successfully applied (e.g., Neumann, 2010; Neumann et al., 2015). An assessment for the model performance in the northern Baltic Sea is given in appendix A.

## 3 Results

### 3.1 Sea ice samples

Bulk salinity of the sea ice cores are summarized in Table 1. Salinity measurements with the salinometer and the mini CTD of the melted ice core water are very close and therefore, the mini CTD measurements appear to be reliable. The mean bulk salinity of our sea ice samples amount to about $0.6\mathrm{g\,kg^{-1}}$ and the sea surface salinity at the ice core stations showed values between $2.95\mathrm{g\,kg^{-1}}$ and $3.0\mathrm{g\,kg^{-1}}$.

The sampled brine volume at station 9 was too small for further analysis. Brine salinity for stations 8 and 10 is listed in Table 2. As for the ice core samples, salinity measurements with mini CTD and salinometer are very close to each other. The mean brine salinity is about $15\mathrm{g\,kg^{-1}}$. Altogether, the sea ice measurements can be summarized as follows: The mean sea ice bulk salinity of our samples is about $0.6\mathrm{g\,kg^{-1}}$. Taking into account a sea surface salinity of $3.0\mathrm{g\,kg^{-1}}$, 2.4g salt are rejected from 1kg frozen sea water into the water column, while the rejected brine shows a salinity of $15\mathrm{g\,kg^{-1}}$.

**Table 1.** Ice core samples from stations 8, 9, and 10. Location shows which part of the core is used for analysis. Top, mid, and bot are the upper, middle and lower part of the ice core, respectively. Bulk is for the whole ice core. Sal.(C) and Sal.(S) are the absolute salinity measured with mini CTD and salinometer, respectively.

| Stat. # | Core # | Location | Sal.(C) [g/kg] | Sal.(S) [g/kg] |
|---|---|---|---|---|
| 8 | 1 | top | 0.81 | 0.86 |
| 8 | 1 | bot | 0.67 | 0.70 |
| 8 | 2 | bulk | 0.54 | 0.77 |
| 8 | 3 | bulk | 0.81 | |
| 8 | 4 | top | 0.93 | 0.80 |
| 8 | 4 | mid | 0.46 | 0.46 |
| 8 | 4 | bot | 0.73 | 0.78 |
| 9 | 2 | mid | 0.80 | 0.77 |
| 9 | 2 | bot | 0.48 | 0.48 |
| 9 | 3 | top | 1.38 | |
| 9 | 3 | mid | 0.60 | |
| 9 | 3 | bot | 0.38 | |
| 9 | 4 | bulk | 0.64 | |
| 10 | 1 | top | 0.82 | |
| 10 | 1 | bot | 0.58 | |
| 10 | 2 | top | 0.53 | |
| 10 | 2 | bot | 0.49 | |
| 10 | 3 | top | 0.73 | 0.71 |
| 10 | 3 | bot | 0.70 | 0.70 |

**Table 2.** Brine samples from stations 8 and 10. Sal.(C) and Sal.(S) are the absolute salinity measured with mini CTD and salinometer, respectively.

| Stat. # | Sal.(C) | Sal.(S) |
|---|---|---|
| 8 | 11.62 | 11.65 |
| 10 | 17.94 | 17.94 |

The spectral absorption of CDOM was measured at samples from three ice stations (8, 9, 10, Fig. 3). Surface water samples from ice holes and CTD rosette were compared with brine and water from different melted ice layers — mainly top, middle and bottom layer. The absorption shows strong differences between ice samples, surface water, and brine. Highest values were measured in brine and lowest values in ice water while only little differences were found within the different groups. Absorption measurements of brine was possible only at station 10 due to low brine volumes sampled at stations 8 and 9. The measured spectra are shown in Fig. 5.

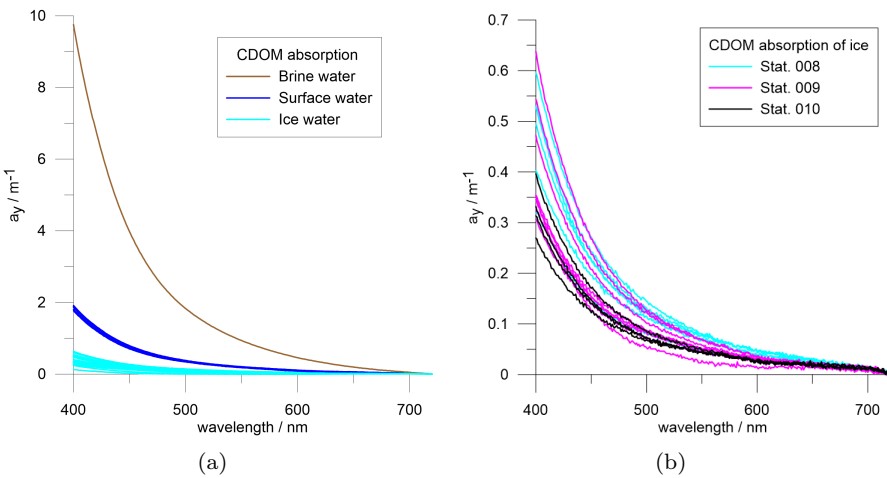

**Figure 5.** Spectral absorption at the sea ice stations including surface water from ice holes and CTD-rosette samples, brine, and water from different melted ice core sections at stations 8, 9, and 10 (Fig. 3). b) shows melted sea ice absorption separately.

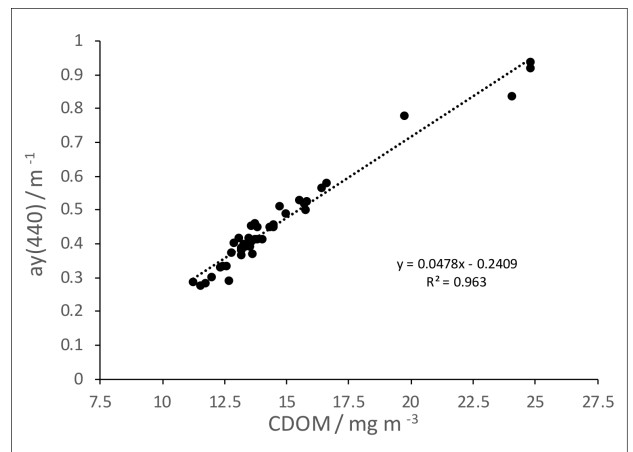

**Figure 6.** Linear regression between CDOM absorption at 440nm and CDOM content derived from Wetlabs CDOM fluorescence sensor.

For the relation between CDOM absorption at 440nm and CDOM content estimated from Wetlabs CDOM fluorescence sensor (chap. 2.2), we derived a linear regression (Fig. 6). This regression can be used to calculate CDOM absorption from CDOM content and vice versa.

CDOM concentrations at station 10, estimated from the relation in Fig. 6, are listed in Table 3. Similar to salt, CDOM concentration increases in brine due to sea ice formation. However, the increase for salt ($17.94 \mathrm{g\,kg^{-1}} : 2.95 \mathrm{g\,kg^{-1}}$) is stronger than the increase for CDOM ($103.99 \mathrm{mg\,m^3} : 23.8 \mathrm{mg\,m^3}$). That means, relatively more CDOM remains in sea ice. Müller et al.

**Table 3.** CDOM absorption and CDOM concentration in brine, water, and sea ice at station 10.

|  | $a_y(440)$ m$^{-1}$ | CDOM mg m$^{-3}$ |
|---|---|---|
| Brine | 4.73 | 103.99 |
| Surface Water | 0.898 | 23.8 |
| Sea Ice (min) | 0.14 | 7.97 |
| Sea Ice (max) | 0.175 | 8.7 |

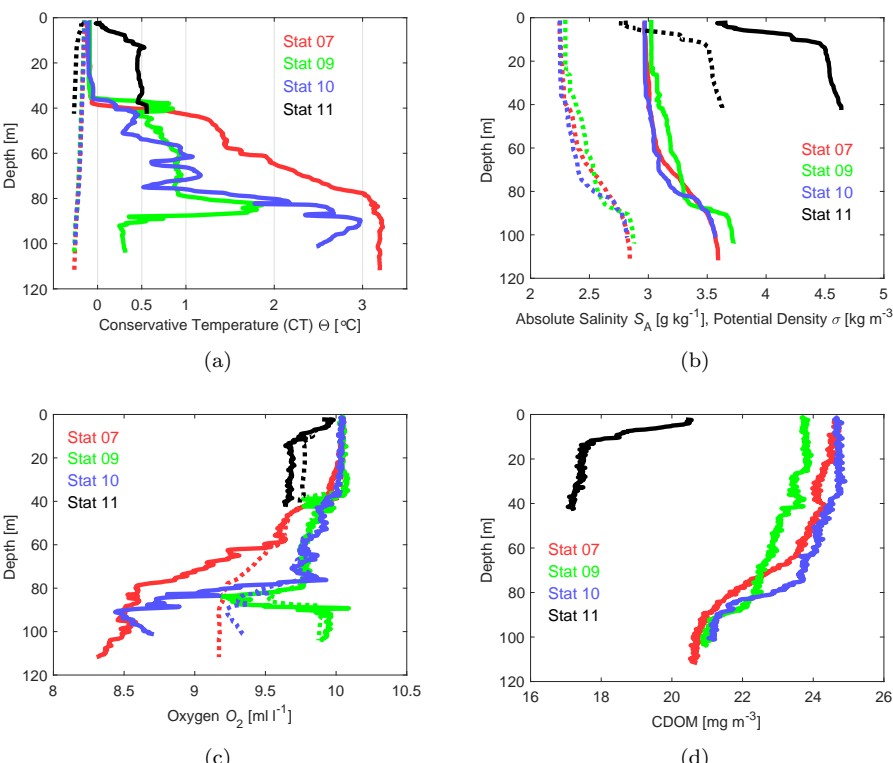

**Figure 7.** Shipborne CTD profiles at stations 7, 9, 10, and 11 (Fig.3) for conservative temperature (a), absolute salinity (b), oxygen (c), and CDOM (d). Dotted lines show the freezing temperature (a), the potential density (b), and the saturation oxygen concentration (c).

(2011, 2013) show that CDOM in Baltic Sea ice is enriched compared to salt with an enrichment factor of up to 39%. Our estimates for station 10 show an enrichment of 50% to 70%.

### 3.2 Water column samples

Figure 7 shows temperature, salinity, oxygen, and CDOM profiles from stations 7, 9, 10, and 11 (see Fig. 3 for locations). In 5 the Bothnian Bay, temperature near the surface is close to the freezing temperature. The density profiles suggest that the upper

20m to 30m are well mixed. At the Northern Quark (station 11), stratification starts just below the surface, presumably due to water from the Bothnian Bay which is continuously mixed with the underlying water from the Bothnian Sea.

A weak stratification also existed below the sea ice measured with the hand-held mini-CTD through the bore holes. Temperature stratification starts at a depth of about 25m. We do not show the data since they are very similar to the data measured with the shipborne CTD.

At all Bothnian Bay stations, temperature and salinity increase with depth below the mixed layer while oxygen is decreasing, either due to salinity and temperature increase (still close to saturation concentration) or due to oxygen consumption. There is an exception at station 9, where a 20m thick layer above the ocean floor is well oxygenated, even slightly over-saturated, showing a very low temperature. We assume that this water mass was formed from water which recently was in contact with the sea surface and therefore, indicate a ventilation event of the bottom water at station 9. To identify a possible genesis of this bottom water, we analyzed the temperature-salinity (TS) properties based on our CTD measurements.

The TS characteristics are shown in Fig. 8. In addition to the Bothnian Bay stations, we included station 12 from the northern Bothnian Sea. Only data points with oxygen concentrations close to saturation and from station 12 only the upper 20m are included. This restriction is justified by the fact that the water mass of interest, the bottom water at station 9, is saturated with oxygen and the Northern Quark sill depth is shallower than 20m and constrains inflows from the Bothnian Sea.

The bottom water from station 9 (well oxygenated) is roughly in the middle of Fig. 8, where mixing lines are crossing. The light greenish, solid lines indicate mixing of water masses from the Bothnian Sea and Bothnian Bay which potentially could have contributed to the bottom water at station 9. The surface water in the Bothnian Bay is close to freezing temperature down to 40m depth (Fig 7) and no candidate for forming the bottom water of station 9. The northern Bothnian Sea surface water is at the other end (right side) of the mixing lines. The greenish dash-dotted line in Fig. 8 is the mixing line, if brine and Bothnian Bay water would form station 9 bottom water. Brine properties have been chosen according to our observations SA=15g kg$^{-1}$ and CT=-0.79°C (freezing temperature). Based on the temperature constraints given by TS characteristics, we suggest that recent Bothnian Bay surface water was not contributing to station 9 bottom water.

The salinity – CDOM relationship is shown in Fig. 9. The main source for CDOM in the northern Baltic Sea are yellow substances carried by rivers into the Baltic Sea. Yellow substances in the Baltic Sea are relatively refractory and therefore, a linear CDOM salinity relation exists (e.g., Harvey et al., 2015). All oxygen saturated water masses are along the mixing line.

### 3.3 Model simulations

The propagation of Bothnian Sea surface tracer is shown in Fig. 10 as the surface tracer concentration just above the sea floor, i.e. former surface water already descended to the bottom. Surface water from the Bothnian Sea arrives at the area of our observations in mid-March. The snapshot from May 31st shows the areas which are ventilated until the end of the model simulation where the dense water plume arrives at the most northern deep parts of the basin.

In Fig.11, we show the time development of the surface tracer concentrations at station 9. A considerable concentration of Bothnian Sea surface water arrives around the 10th of March in the bottom water of station 9. Both surface tracer, from the Bothnian Bay and the Bothnian Sea, account for 50%–60% of the bottom water. The contribution of brine is less than 1% and

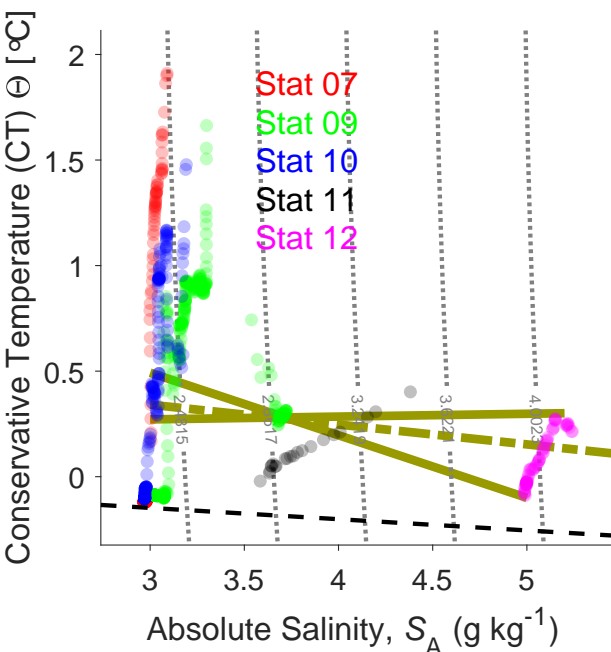

**Figure 8.** TS diagram for stations 7, 9, 10, 11, and 12 (Fig. 3). Gray dotted lines show the density. Greenish solid and dash-dotted lines show mixing lines for station 9 bottom water from Bothnian Bay and Bothnian Sea surface water, and from Bothnian Bay water and brine, respectively. Brine is outside the figure with SA=15g kg$^{-1}$ and CT=-0.79°C. Only data points with oxygen concentration close to saturation are considered. For station 12, data points are restricted to a depth shallower than 20m. The dashed black line is the freezing temperature. Stronger opacity of the data points refers to a higher number of data.

negligible. The mean age of surface water is shown in the lower panel of Fig 11. In the beginning of March, it shows the age of the small amount of surface water arrived shortly after the simulation have been started. The mean age decreases by 20–25 days when the pulse of surface water (March 7th–10th) arrives. The last surface contact of Bothnian Sea water was about 35 days before, i.e. around February 1st. At this time, surface water masses from the Bothnian Sea and Bothnian Bay have been mixed.

5    The resulting density initiated a down-slope transport into the Bothnian Sea from the surface.

## 4    Discussion

In March 2017, a well oxygenated and cold bottom water layer was observed at station 9 in the center of the Bothnian Bay (Fig. 7). The water column showed a pronounced density stratification mainly due to a halocline in about 80m depth with less oxygen than in the bottom water. Therefore, it is very likely that the oxygen rich water arrived at this position rather by

10    lateral intrusion or inflow than by vertical mixing. In the sea-ice-covered Bothnian Bay, two mechanisms potentially could have produced the observed water mass. (i) Bothnian Sea surface water crossed the Northern Quark, was mixed with Bothnian

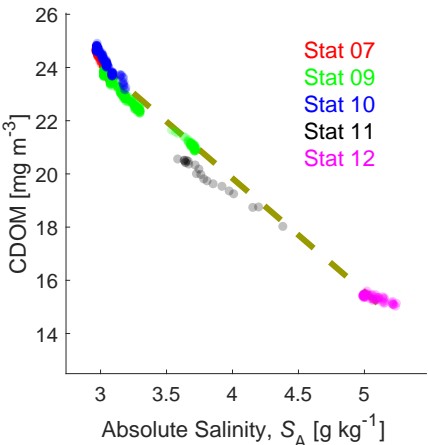

**Figure 9.** Salinity – CDOM diagram. The dashed line is the mixing line. Stronger opacity of the data points refers to a higher number of data. Data of station 7 are mostly covered by station 9 and 10 data.

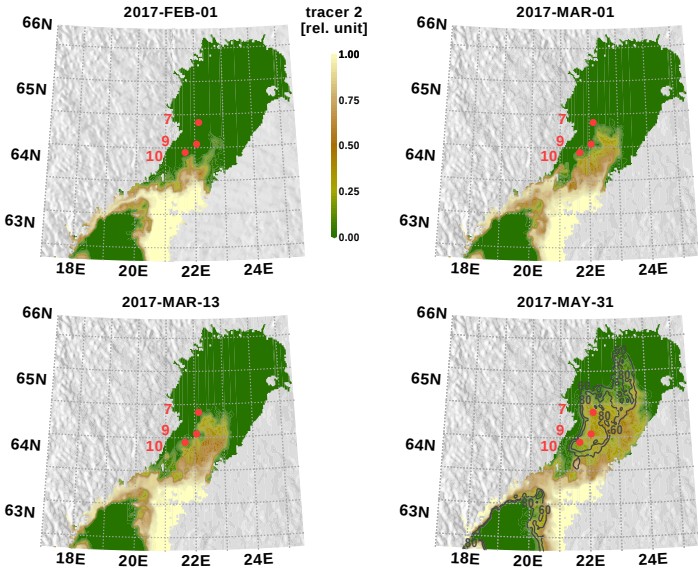

**Figure 10.** Bothnian Sea surface tracer concentration in the bottom model layer. Red dots indicate the observation stations. In the fourth sub-figure, bathymetry contour lines for 60m and 80m depth are shown.

Bay surface water, descended to the bottom due to high density and followed the topography into the basin, or (ii) brine release from sea ice and ambient water formed a denser water mass, preferentially in shallower coastal areas which then, as a gravity current, arrived at the deep basin.

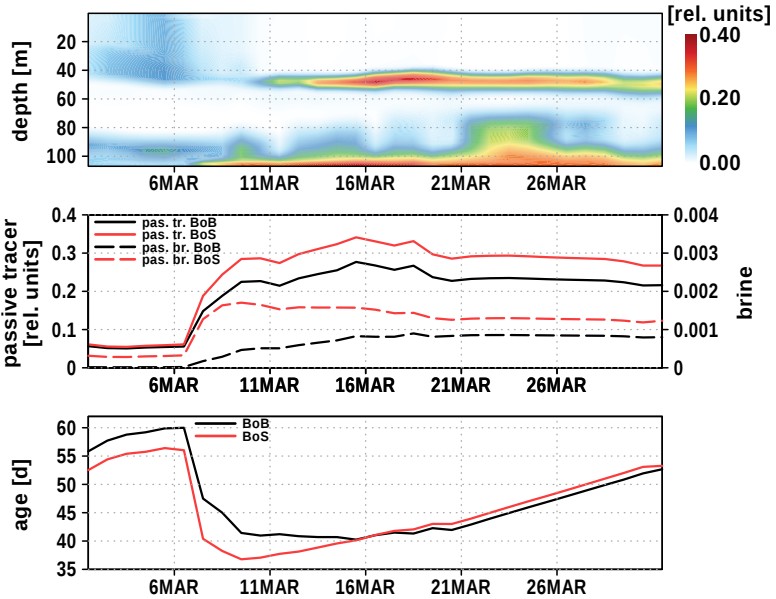

**Figure 11.** Time series of Bothinan Sea and Bay surface tracer at station 9 (Fig. 3). The upper panel shows the vertical distribution of Bothnian Sea surface water; the middle panel shows the concentrations of Bothnian Sea (red) and Bay (black) surface water, and the concentration of brine (dashed, right y-axis) in the bottom water at station 9; the lower panel shows the mean age of Bothnian Sea and Bay surface water in the bottom water at station 9.

The low temperature and the very high oxygen concentration (Fig. 7) reveal that the contributing water masses were in contact with the surface recently. A TS analysis in Fig. 8 suggest that mixing of the observed Bothnian Sea (magenta dots) and Bothnian Bay water hardly can result in the observed oxygen rich and cold bottom water. Only a small portion of Bothnian Bay water in about 40–50m depth (blue dots) shows the appropriate TS properties. However, assuming temperatures would be

higher earlier in the winter season, surface water from the Bothnian Sea and from the Bothnian Bay can mix into the observed bottom water.

The model simulation with a passive tracer approach supports these findings. The majority of new bottom water is formed from surface water which starts to descend in the beginning of February.

We show in Fig. 12 the sea surface temperature (SST) development as seen in our model simulation. The shown SST is an

area average for the southern Bothnian Bay and the northern Bothnian Sea. In both areas, the SST simultaneously decreases and a suitable SST, forming the observed bottom water, can be found in the beginning of February.

However, app. 30% of simulated bottom water was not in contact with the surface. This is also evident from the simulated temperature. It drops from 2.8°C to 1.6°C with the arrival of the new water mass and reflects that about one third is from older and warmer, intermediate or bottom water. The overestimation of entrainment is a known issue of z-coordinate models,

especially for down flowing dense water plumes (Winton et al., 1998).

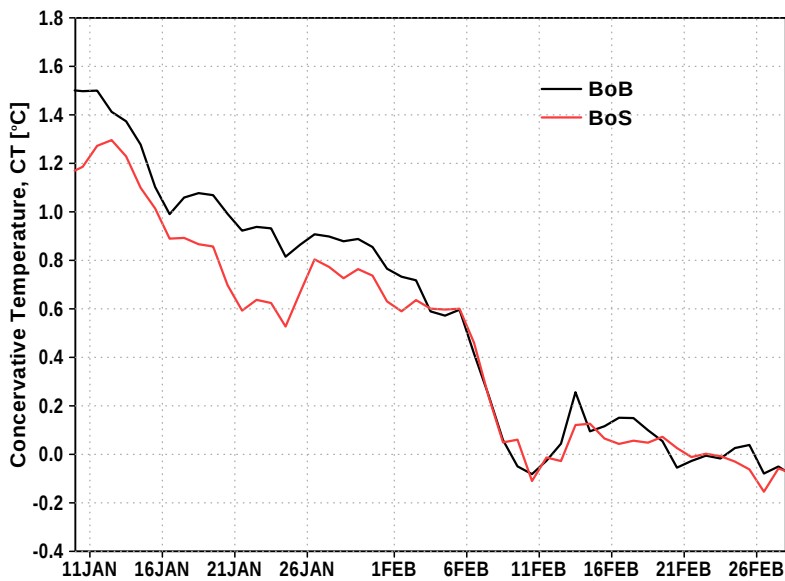

**Figure 12.** Model SST in the southern Bothnian Bay (black) and the northern Bothnian Sea (red).

Harvey et al. (2015) show a linear relationship between CDOM fluorescence and CDOM absorption, and CDOM absorption and salinity (negative slope) in the Bothnian Sea, resulting in a linear relationship (negative slope) between CDOM fluorescence and salinity. Figure 9 clearly shows this relationship. Assuming that CDOM (yellow substances) are largely conservative, the bottom water at station 9 could be the result of mixing Bothnian Sea and Bothnian Bay water. For an analysis of water masses,

we estimate the mixing ratio mx. mx is the volume ratio of Bothnian Bay surface water $S_{BS}$ to Bothnian Sea surface water $S_Q$ giving Bothnian Bay bottom water $S_{BB}$ based on measured salinity.

$$mx = \frac{S_Q - S_{BB}}{S_{BB} - S_{BS}}$$

With the salinity of the surface water at station 12 $S_Q = 4.9$, of Bothnian Bay surface water $S_{BS} = 3.0$, and of Bothnian Bay bottom water at station 9 $S_{BB} = 3.75$, mx is 1.53. Using this mixing ratio, we can estimate the CDOM concentration in

the Bothnian Bay bottom water: Given surface CDOM concentrations in the Bothnian Sea ($15.4\,\mathrm{mg\,m^{-3}}$) and Bothnian Bay ($24\,\mathrm{mg\,m^{-3}}$) result in a bottom water concentration in the Bothnian Bay of $20.6\,\mathrm{mg\,m^{-3}}$. This value is close to the observed value of $21\,\mathrm{mg\,m^{-3}}$ (Fig. 7).

Mixing of brine and Bothnian Bay surface water cannot decrease CDOM to the observed level. If CDOM behaves similar as salt during freezing, that is, concentration increases in brine, the mixed water would show an elevated CDOM concentration

like salt. Müller et al. (2011) and Müller et al. (2013) show that CDOM is enriched in sea ice compared to salt. The enrichment factor of CDOM in sea ice relative to salt is in the order of 1.3. Therefore, the CDOM concentration in brine is less increased than salinity but still higher than in the surface water and a brine induced bottom water would have higher CDOM concentra-

tions than the surface water. These findings agree with our model simulations showing virtually no brine in the bottom water (Fig. 11).

Model simulations and observations show that bottom water ventilation in the Bothnian Sea is due to mixing of surface water at the Northern Quark forming dense water which then initiates a down-flow cascade into the Bothnian Bay. Shapiro et al. (2003) developed a theory for dense water cascades at continental shelves and derived a criterion discriminating between an accelerated event and a steady Ekman layer flow. The conditions for an accelerated cascade (Shapiro et al., 2003, eq. 9) are not fulfilled in the case of the Bothnian Bay, thus the down-slope flow is rather a bottom boundary layer Ekman flow than an accelerated event.

A density estimate based on our observed data show that the density gradient between Bothnian Bay bottom water and surface water at the Northern Quark vanishes if the SST exceeds 8°C. Therefore, ventilation will occur in winter only.

## 5   Conclusions and Summary

During a cruise in March 2017, we took sea ice samples in the northern Baltic Sea, the Bothnian Bay. The bulk ice salinity was about $0.6\mathrm{g\,kg^{-1}}$. Brine samples showed a salinity of about $15\mathrm{g\,kg^{-1}}$ and the surface water a salinity of about $3\mathrm{g\,kg^{-1}}$. In addition, CTD casts were performed at 3 different locations. At one location, the bottom water was saturated with oxygen and the temperature was very low (0.3°C) compared to the other two stations. We find that this bottom water was formed due to a recent intrusion of former surface water. Complementing the observations, a numerical model experiment has been designed allowing to track different water masses.

Observations, and especially CDOM data, exclude a mechanism where brine contributes to dense water formation. Also the model simulation gives no evidence for a contribution of brine.

The plausible ventilation mechanism is the inflow of Bothnian Sea surface water into the Bothnian Bay. We found a possible time for such an inflow 5 weeks before our measurements. This mechanism can only work if the sea surface salinity of the adjacent basin is considerably higher than the salinity of the bottom water. A prerequisite for this configuration is a shallow sill separating the basins. The depth have to be sufficient shallower than the halocline to prevent saline deep water inflows. A necessary horizontal density gradient is established only for a low surface water temperature. Consequently, these ventilation events occur in the winter season. These findings confirm Marmefelt and Omstedt (1993) who exclude haline convection for the Bothnian Sea based on budget estimates and considered this process as very unlikely for the Bothnian Bay.

We observed the ventilation at one station only. However, the model simulation shows that the dense water plume progresses further northwards during spring and ventilates also the northernmost part. We are aware that our observations are a snapshot supported by numerical model simulations. Therefore, we are planning an extended campaign including moorings equipped with current meters. The model simulations are an excellent tool stimulating and guiding CTD and mooring stations.

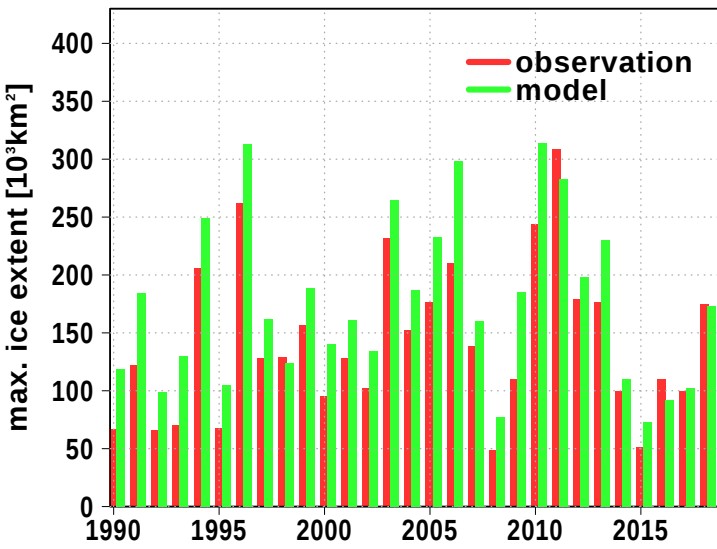

**Figure A1.** Simulated (green) and observed (red) maximum sea ice extent. Observations are obtained freely from https://en.ilmatieteenlaitos.fi/ice-winter-in-the-baltic-sea.

*Data availability.* All sea ice and brine data are in this text. Data from the hand-held CTD and shipborne CTD data are available from http://doi.io-warnemuende.de/10.12754/data-2019-0002. Model simulation data are freely available from https://thredds-iow.io-warnemuende.de/thredds/c THREDDS-Baltic_ref_2020-04-21-10.

## Appendix A: Model assessment

5  In this section, we demonstrate the model performance for reproducing sea ice, temperature, and salinity in the northern Baltic Sea. Although the model run starts in 1948, we show the validation from 1990 to 2018, the last 30 years of the simulation. We skip the first 40 years, because the model was still drifting due to the long residence time of the Baltic Sea, especially in the northern parts.

Figure A1 shows the annual maximum ice extent from observations (red bars) and from the model simulation (green bars).
10  The variability is well reproduced by the model while in most years the extent is somewhat overestimated. In the model, we summed up all ice classes including frazil which might not always be part of the observations.

In the following figures, we show temperature and salinity data from two stations in the Bothnian Bay and in the Bothnian Sea, respectively. We focus on surface data (green) and data from close to the sea floor (blue). All observations are freely distributed by ICES (www.ices.dk). In Fig. A2, the locations of the two stations are shown.

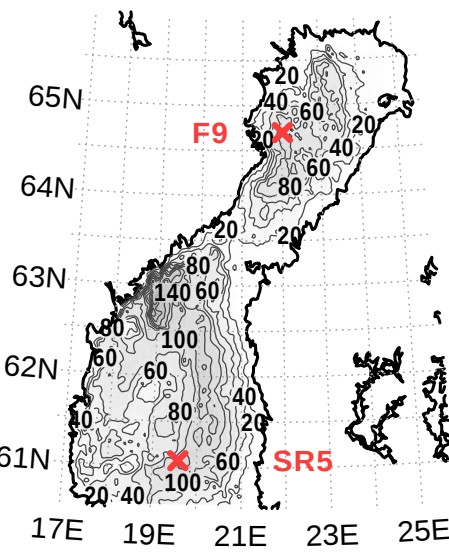

**Figure A2.** Location of the validation stations. The map was created using the software package GrADS 2.1.1.b0 (http://cola.gmu.edu/grads/), using published bathymetry data (Seifert et al., 2008).

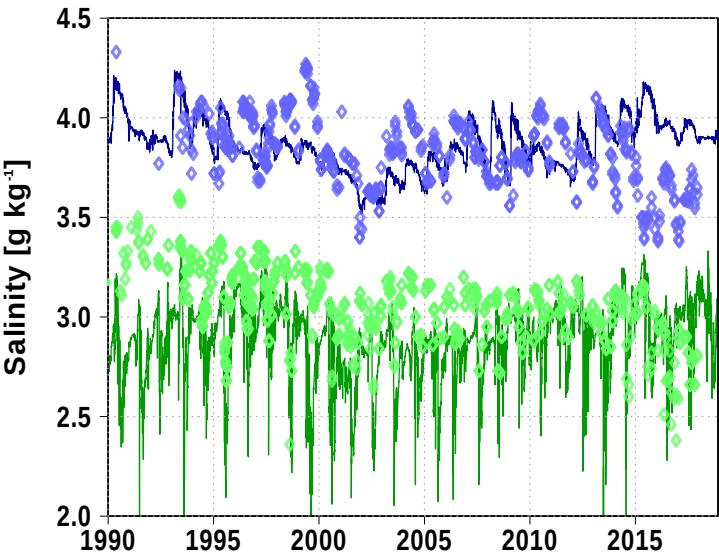

**Figure A3.** Surface (green) and bottom (blue) salinity at station F9. Solid line: Model, Diamonds: Observations (www.ices.dk).

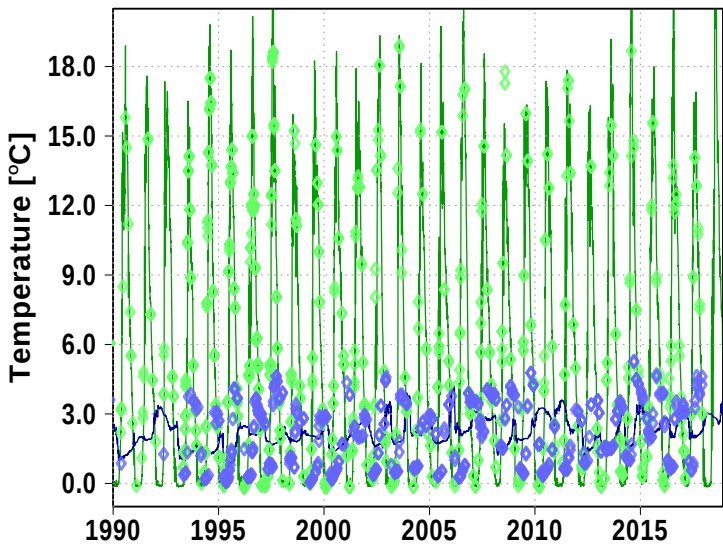

**Figure A4.** Surface (green) and bottom (blue) temperature at station F9. Solid line: Model, Diamonds: Observations (www.ices.dk).

Temperature and salinity are reasonably reproduced in the Bothnian Bay (F9). A weakness are the low bottom temperature events which are not reproduced well by the model. The reason is the overestimation of entrainment in z-coordinate model as discussed in section 4.

Salinity in the Bothnian Sea (SR5) bottom water is overestimated by app. $0.5 \mathrm{g\,kg}^{-1}$. Surface salinity and temperature show a reasonably good performance.

In summary, we assess that the model is able to reproduce the relevant processes sufficiently. The experiment with the passive tracers show that despite the enhanced entrainment also the ventilation of bottom water is reflected in the model.

*Author contributions.* TN and MM designed the experiment, TN designed and performed the model simulations, HS and MG performed optical measurements, MK and DN performed CTD and ice core measurements. All authors contributed to data analysis and writing the manuscript.

*Competing interests.* The authors declare that they have no conflict of interest.

*Acknowledgements.* We are very grateful for the assistance of the crew members of the RV *Maria S. Merian* who made possible research in the sea-ice-covered Baltic Sea. Chief scientist Prof. R. Schneider always supported our experiments, especially the sea ice coring. Ingo

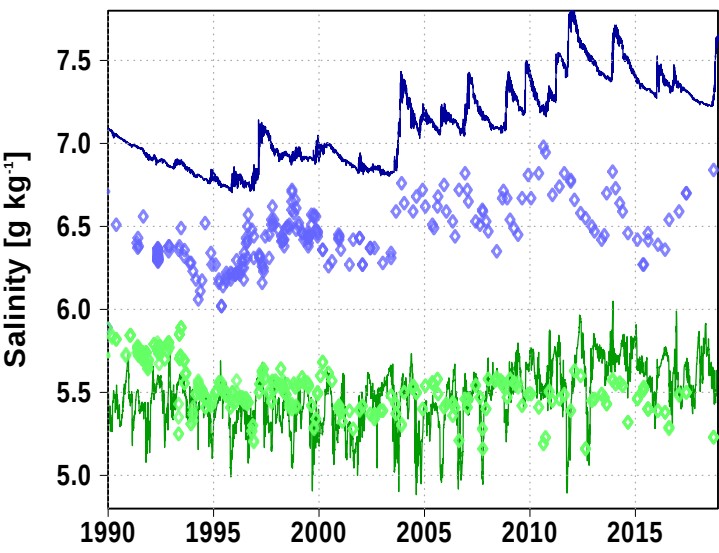

**Figure A5.** Surface (green) and bottom (blue) salinity at station SR5. Solid line: Model, Diamonds: Observations (www.ices.dk).

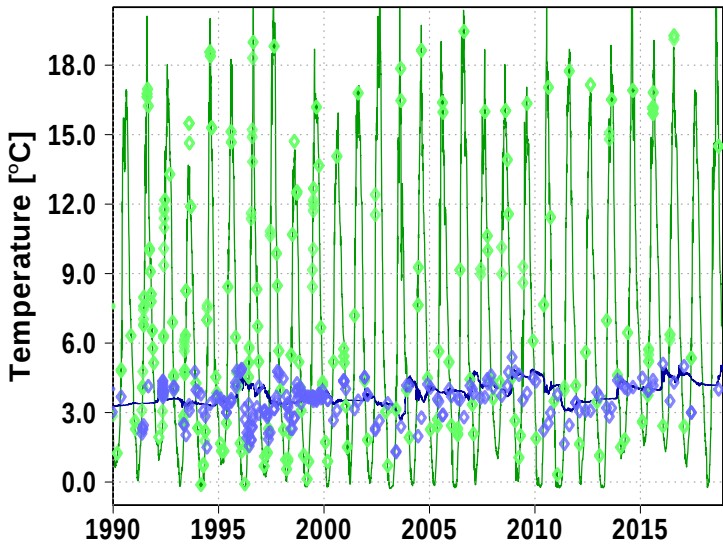

**Figure A6.** Surface (green) and bottom (blue) temperature at station SR5. Solid line: Model, Diamonds: Observations (www.ices.dk).

Schuffenhauer operated the CTD and the salinometer. Computational power was provided by the North-German Supercomputing Alliance (HLRN). We are very grateful for three anonymous referees who helped considerably improving the manuscript.

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
