# Peer review of "Ventilation of the Northern Baltic Sea"

_Ocean Science, 2019_

## Referee Comment (RC1) · Anonymous Referee #1 · 23 Aug 2019

The paper reports recent observation-based finding of the ventilation event in the bottom water of the Bothnian Bay in March, 2017. The authors conclude that the ventilation is most probably happened due to mixing of Bothnian Sea and Bothnian Bay surface water which resulted in formation of dense water, which was able to replace the older bottom water.

General assesment Unfortunately, I cannot recommend this manuscript in the present shape for publishing. Major revision is required. Although, the theme is interesting and the narrative is logical, there is a problem with the message, the authors are trying to deliver. As a matter of fact, they could not support by their data any of the hypotheses, which could explain the origin of the highly-oxygenated bottom water at the solitary station 19. In the Conclusion section the authors agree that the ventilation mechanism, they suggested is highly speculative, since it is not confirmed strongly by any independent data at hand. There are also several questions to the text, which are listed

below.

Questions/comments

Abstract. The abstract is not sufficiently informative. Basically, there is just one sentence, which describes, what this paper is about. I would suggest to exclude all general wording about the Baltic Sea, but to add more about the essence of this paper.

Page 2, lines 2-3: I would suggest to add more detailed explanation (not just reference on Peterson, 2018), how melting sea ice can produce considerable salt fluxes into the ocean? This is quite a new knowledge, and it is important to explain it in more details.

Page 2, lines 3-4: The references to Aagaard,et al., (1981) and Skogseth et al. (2008) are not relevant in this context. These papers describe specific events of dense water formation and do not consider general theory of this event at all. Taking into account that the authors are considering horizontal advection, as the most probable mechanism of the surface water densification in the Bothnian Bay, I would suggest them to read papers by Shapiro et al (2003) and Ivanov et al. (2004), which summarized all known mechanisms of dense water formation and cascading (not only in the Arctic, but worldwide).

Page 15, lines 19-20: "In areas shallower than the pycnocline, dense water can accumulate at the sea floor and form density driven plumes guided by topography"

This is very speculative statement, which is not confirmed by the provided measurements, but only by the references to the older studies, where this idea was also rather claimed, but not strongly supported.

---

## Referee Comment (RC2) · Anonymous Referee #2 · 24 Sep 2019

This discussion paper presents ocean and sea ice observations from a cruise to the Northern Baltic Sea.

While there is nothing controversial and the discussion and conclusions are somewhat based on the observations, there is a quite modest set of findings. Very little evidence can be provided for how the ventilation is actually taking place. There is one profile that shows ventilated water, and very limited testing of mechanisms or modelling. As it stands now there is no use for all the sea ice observations, as the authors are suggesting the ventilation happens through advection.

The paper is in my view below the minimum of what should be published in an international scientific journal, and appears at present like a cruise report. Until further observations or simulations can be provided, it does not appear like a proper scientific paper. The oceanographic handwork done is of a good quality though, but the authors

have drawn the conclusions based on a very limited set of observations.

The paper largely ignores earlier literature on Baltic sea in special, and on the other relevant processes in question.

There are in my view a number of substantial improvements to be made if the paper should be accepted beyond the "Discussion" part of the journal – as outlined below.

General comments:

- The layer close to bottom at one station (station 9) is marginally interesting. Clearly one would have wanted more than one station to show the persistence of any suggested process. This water is high in oxygen, but also 0.5 deg C above freezing. This indicates that it is not related to sea ice formation, or brine driven convection. So why is all the sea ice observations included? Really – it has no use as the paper is written at the moment. Obviously the authors would have liked to find evidence for the "brine hypothesis" – but they have not.

- There is generally a small number of citations given. While it is good practice not to overflow with too many, here it is on the sparse side. And one suspects that the authors have spent a limited effort on finding relevant studies. A good example is for the experimental studies in polynyas (Page 2, line 5). Clearly there are many more observations available from polynyas, both in the Arctic and Antarctica. As noted by the other reviewer are also some literature on the down-flow required. Examples on earlier polynya studies are given at the end. In general is there also much more available studies of Baltic sea ice available, where the few seas ice samples could be compared to.

Specific Comments:

Page 2, line 7. The Arcic is a name, should always be spelled with capital A. Correct throughout.

Page 3, line 6: Use of "Fast ice" is wrong. Fast ice means sea ice frozen onto the

shore. Here I think you mean pack ice?

https://www.jcomm.info/index.php?option=com_oe&task=viewDocumentRecord&docID=14598

Page 3, line 11 It is not clear how you sample the brine. You state that:" holes approximately half the depth of the ice thickness were drilled to collect brine". Do you mean that you take out the core, and wait for water to drain back into the whole? How do you know this is the brine? The brine salinity is very tricky to sample, and conditions here are very special with the super low surface salinities.

Page 4 – lower 4 lines. You simply state you used the (standard) Guildline's Autosal 8400B and the accuracy. It is a standard procedure in the field.

Page 4 –line 8: Are you sure you closed the bottles on the way down? With higher pressures this would lead to the bottles imploding, so the standard is to do this on the way up.

Page 6 –line 7: What do you mean by; "CTD and salinometer measurements of the melted ice core water are very close and, therefore, the CTD measurements appear to be reliable"? How can you take a CTD measurement of the melted ice core? A CTD needs to be fully submerged in ocean water to work, and measures the conductivity over a much larger volume of water that is inside the conductivity cell. . .

Page 6 – line 12; "The mean sea ice bulk salinity in the Bothnian Sea is about 0.6g kg-1". This is a very strong claim when you have ice cores from 3 locations. . ...

Page 8, Figure 4 caption: ice sheets – this means the large piece of ice on Greenland and Antarctica. You may mean "ice core"?

Page 10. Figure 7. The mini CTD observations appear close to the ship-born CTD. If they are plotted in the same figure – then one could see if there are any differences – but this appears not to be the case. This figure is not valuable – unless there are some significant differences – and then these should be shown in Figure 6.

Figure 8: Is this the ship CTD data? Why then is not the warmest water on Station 10 and 12 about +3 deg C visible? And

Page 12 – line 10. Please use one temperature throughout a paper. It is fine to use the new conservative temperature, but then you should use it throughout.

Page 13 – line 1: "we do not have information on surface salinity or currents." This is exactly the main problem. Very little data is available, and then one cannot really conclude on the suggested processes either. A numerical model could have amended this in a nice way.

Page 13 – line 6: Also here; "there are some indications that surface water from the Bothnian Sea have been mixed with Bothnian Bay water forming the observed bottom water at station 9". Some anecdotal indications are not really enough to claim that one has new findings worth publishing in an international journal.

Page 14, line 11-16: While I am no expert in biological processes it is clearly possible that there is growth of organic mater in sea ice, and this should be discussed. A fairly new paper (Assmy et al 2017) also finds that phytoplankton can also grow below a snow cover.

Page 14, line 18: polyniyas is spelled wrongly.

Page 14, line 17 – Page 15, line 7. While this is possible in the Bothnian Bay – you do not have any observations that indicate that this is going on. IF you added some simulations that this is likely, then this text could remain – otherwise it should be deleted.

Page 15, line 8 – 20. This section finally contains some calculations about the brine water "hypothesis". The calculations appear OK - but does not use a proper range in forcing and boundary conditions. How representative is the 0.2 m of ice thickness? Is there any freshwater discharge during winter?

Suggested citations – there are many more:

Assmy et al (2017) Leads in Arctic pack ice enable early phytoplankton blooms below snow-covered sea ice, Scientific Reports, 7:40850, 2017, doi: 10.1038/srep40850

Cavalieri, D. J., and S. Martin (1994), The contribution of Alaskan, Siberian, and Canadian coastal polynyas to the cold halocline layer of the Arctic Ocean, J. Geophys. Res., 99, 18,343– 18,362.

Gordon, A. L. (2014), 'Southern Ocean polynya', Nature Climate Change 4(April), 249– 250.

Winsor, P., and Björk, G. ( 2000), Polynya activity in the Arctic Ocean from 1958 to 1997, J. Geophys. Res., 105( C4), 8789– 8803, doi:10.1029/1999JC900305.

Humfrey Melling, Yves Gratton & Grant Ingram (2001) Ocean circulation within the North Water polynya of Baffin Bay, Atmosphere-Ocean, 39:3, 301-325, DOI: 10.1080/07055900.2001.9649683

Granskog et al (2006) Sea ice in the Baltic Sea – A review, Estuarine, Coastal and Shelf Science, Volume 70, Issues 1–2

Granskog et al (2005). Characteristics and potential impacts of under-ice river plumes in the seasonally ice-covered Bothnian Bay (Baltic Sea). Journal of Marine Systems 53,

---

## Referee Comment (RC3) · Anonymous Referee #3 · 30 Sep 2019

This paper tries to justify that oxygen rich bottom water found in one profile collected in the Bothnian Bay, may have been formed by inflowing water from the Bothnian Sea mixed with surface water in the Northern Quark, and not by salt release from sea ice. The data set is very small, and the processing done poorly explained. It is also hard from the discussion to grasp that the above explanation is what the authors want to say. To make this manuscript more readable they should state more clearly in the discussion whether each explanation they try own ends up with a plausible explanation. I do not suggest any places where to do this in particular, but both the abstract and discussion and conclusions should become clearer. Perhaps add your hypotheses at the end of your introduction. Than it would be easier to state of your data support or do not support each of the hypotheses.

Page 3, line 12: Explain how you collected the brine.

[Figure]

Page 4, Figure 3 caption: at (red crosses) to stations 7-10.

Page 4, lines 1-2: Explain more elaborate how you measured salinity in both ice core samples (Table 1) and brine samples (Table 2). For instance how big did the samples have to be to measure salinity with the CTD. Also, with the low number of samples you collected, why not measure all with the Guildline?

Page 4, lines 2-7: The Guildline Autosal is a standard instrument used with a standard procedure, so this procedure does not need elaborate description. How you collected ice samples and brine and measured their salinity is on the other hand not standard procedure and needs better description.

Page 5, line 10: Justify how you can assume that 'upper 5m are well mixed and saturated with oxygen'. This might be ok for late winter, although I do not have any reference to recommend.

Page 6, lines 13-14: Can you justify the assumption that all the rejected salt is trapped in brine pockets inside the ice? Some of it can be released into the water column.

Page 6, line 14: You can hardly regard an average of two brine samples an average (14.7g/kg). At least, remove the decimal.

Page 7, line 2: from where and to where is the water 'out-flowing'. I would find the term 'in-flowing' more appropriate if it flows from the Bothnian Sea into the Bothnian Bay.

Page 7, line 2, last words: Change to 'A weak stratification'

Page 9 and 10, Figures 6 and 7: Try to use colors that are more easy to separate from each other. Especially Stations 9 and 10 with purple colors.

Page 9, line 6: Mixing lines do not 'show' water masses. They indicate along which line a mix between two source water types can be placed.

Page 10, Figure 8: It is confusing when the end points of the mixing lines go beyond the source water masses. It is clear where the end water mass is, but not the source water

masses. You should also indicate better which 'greenish dashed and dash-dotted' lines you are referring to in each case. Where is the brine in this figure, having which temperature? Etc.

Page 11, Figure 9: in this figure, the colors of stations 10 and 12 are difficult to distinguish.

Page 11, line 3: The brine must be way beyond the axis in Figure 8. Again, which TS characteristics do you assume in the brine?

Page 11, line 15: 'mechanisms'

Page 13, line 5: 'weakening'

Page 14, line 18: 'polynyas'

Page 15: make it clearer which formation process you trust and which you do not trust.

---

## Author Comment (AC1) · 26 Nov 2019

First of all, we would like to thank the anonymous referee for the thorough review of our manuscript.

The common concern of all referees is that the amount of data collected during a cruise in the sea ice covered northern Baltic Sea is not sufficient to justify any of the hypotheses raised in the manuscript. Indeed, only three stations could be explored and we hardly can increase the number of in-situ observations. However, the referees proposed an option to save the manuscript by a complementary modeling approach.

We decided to follow this line and will perform a model simulation for the winter season 2016/2017. We will set up a model for the Baltic Sea with earmarked water masses allowing us to identify the origin of water which eventually arrives in the deep water of the Bothnian Bay. Nevertheless, we want to stress that recent ocean models are

not able to reproduce a possible haline convection due to brine release. However, this approach will prolong a revised version of our manuscript and will include substantial changes.

In the following, we respond to the referee's specific remarks. Remarks are shown and followed by our response.

Review #1:

1.) Abstract. The abstract is not sufficiently informative. Basically, there is just one sentence, which describes, what this paper is about. I would suggest to exclude all general wording about the Baltic Sea, but to add more about the essence of this paper.

We will follow the suggestion and update the abstract in the revised version when the model simulations have been completed and evaluated.

2.) Page 2, lines 2-3: I would suggest to add more detailed explanation (not just reference on Peterson, 2018), how melting sea ice can produce considerable salt fluxes into the ocean? This is quite a new knowledge, and it is important to explain it in more details.

An extended explanation will be done in the revised version.

3.) Page 2, lines 3-4: The references to Aagaard,et al., (1981) and Skogseth et al. (2008) are not relevant in this context. These papers describe specific events of dense water formation and do not consider general theory of this event at all. Taking into account that the authors are considering horizontal advection, as the most probable mechanism of the surface water densification in the Bothnian Bay, I would suggest them to read papers by Shapiro et al (2003) and Ivanov et al. (2004), which summarized all known mechanisms of dense water formation and cascading (not only in the Arctic, but worldwide).

We are grateful for the literature recommendation.

4.) Page 15, lines 19-20: "In areas shallower than the pycnocline, dense water can accumulate at the sea floor and form density driven plumes guided by topography" This is very speculative statement, which is not confirmed by the provided measurements, but only by the references to the older studies, where this idea was also rather claimed, but not strongly supported.

The referee is right. In a revised version, we will formulate this statement as a hypothesis which cannot be rejected a priori.

---

## Author Comment (AC2) · 26 Nov 2019

First of all, we would like to thank the anonymous referee for the thorough review of our manuscript.

The common concern of all referees is that the amount of data collected during a cruise in the sea ice covered northern Baltic Sea is not sufficient to justify any of the hypotheses raised in the manuscript. Indeed, only three stations could be explored and we hardly can increase the number of in-situ observations. However, the referees proposed an option to save the manuscript by a complementary modeling approach.

We decided to follow this line and will perform a model simulation for the winter season 2016/2017. We will set up a model for the Baltic Sea with earmarked water masses allowing us to identify the origin of water which eventually arrives in the deep water of the Bothnian Bay. Nevertheless, we want to stress that recent ocean models are

not able to reproduce a possible haline convection due to brine release. However, this approach will prolong a revised version of our manuscript and will include substantial changes.

In the following, we respond to the referee's specific remarks. Remarks are shown and followed by our response.

Review #2:

1.) The layer close to bottom at one station (station 9) is marginally interesting. Clearly one would have wanted more than one station to show the persistence of any suggested process. This water is high in oxygen, but also 0.5 deg C above freezing. This indicates that it is not related to sea ice formation, or brine driven convection.

Here we disagree with the referee. When brine mixes with surface water and vertically subsides, it preserves most probably the surface temperature. If the new water mass then horizontally spreads or moves downslope, it mixes with warmer deep water.

2.) So why is all the sea ice observations included? Really – it has no use as the paper is written at the moment. Obviously the authors would have liked to find evidence for the "brine hypothesis" – but they have not.

Yes, this was one of the motivations for the sea ice observations. We wanted to test the hypothesis whether brine contributes to deep water formation in the northern Baltic or not. We will formulate the hypotheses more pronounced in the revised manuscript.

3.) There is generally a small number of citations given. While it is good practice not to overflow with too many, here it is on the sparse side. And one suspects that the authors have spent a limited effort on finding relevant studies. A good example is for the experimental studies in polynyas (Page 2, line 5). Clearly there are many more observations available from polynyas, both in the Arctic and Antarctica. As noted by the other reviewer are also some literature on the down-flow required. Examples on earlier polynya studies are given at the end. In general is there also much more

available studies of Baltic sea ice available, where the few seas ice samples could be compared to.

We will increase the number of relevant literature.

Specific Comments:

4.) Page 2, line 7. The Arcic is a name, should always be spelled with capital A. Correct throughout.

Will be done in a revised version.

5.) Page 3, line 6: Use of "Fast ice" is wrong. Fast ice means sea ice frozen onto the shore. Here I think you mean pack ice? https://www.jcomm.info/index.php?option=com_oe&task=viewDocumentRecord&docID=14598

Pack ice is the right term.

6.) Page 3, line 11 It is not clear how you sample the brine. You state that:" holes approximately half the depth of the ice thickness were drilled to collect brine". Do you mean that you take out the core, and wait for water to drain back into the whole? How do you know this is the brine? The brine salinity is very tricky to sample, and conditions here are very special with the super low surface salinities.

The brine sampling was like the referee assume. We will describe the sampling procedure more explicitly. We are quite sure that we sampled brine. What else could have caused the high, measured salinity? Of course, the sample could be polluted by snow. However, we took care to remove snow from the vicinity of the sampling station.

7.) Page 4 – lower 4 lines. You simply state you used the (standard) Guildline's Autosal 8400B and the accuracy. It is a standard procedure in the field.

Thanks for this hint.

8.) Page 4 –line 8: Are you sure you closed the bottles on the way down? With higher
pressures this would lead to the bottles imploding, so the standard is to do this on the way up.

Yes, we are sure; exception is the surface bottle because of possible air bubbles. Doing so, we prevent samples from the wake of the down cast. Pressure differences are not so large in the shallow Baltic Sea.

9.) Page 6 –line 7: What do you mean by; "CTD and salinometer measurements of the melted ice core water are very close and, therefore, the CTD measurements appear to be reliable"? How can you take a CTD measurement of the melted ice core? A CTD needs to be fully submerged in ocean water to work, and measures the conductivity over a much larger volume of water that is inside the conductivity cell. . .

We used the handheld (mini) CTD for these measurements. The conductivity cell is very small compared to the sample water volume. We will describe the procedure more explicitly.

10.) Page 6 – line 12; "The mean sea ice bulk salinity in the Bothnian Sea is about 0.6g kg-1". This is a very strong claim when you have ice cores from 3 locations. . ...

We will reformulate this statement: - mean sea ice bulk salinity of the sampled cores -

11.) Page 8, Figure 4 caption: ice sheets – this means the large piece of ice on Greenland and Antarctica. You may mean "ice core"?

We will call it "ice core slices" to prevent misunderstanding.

12.) Page 10. Figure 7. The mini CTD observations appear close to the ship-born CTD. If they are plotted in the same figure – then one could see if there are any differences – but this appears not to be the case. This figure is not valuable – unless there are some significant differences – and then these should be shown in Figure 6.

We will check whether there are any differences. The measurements are not exactly from the same location. Mini CTD observations are directly from below the sea ice

while ship-borne observations are from open water and the surface layer could be disturbed by the ship itself. If there are no differences, Fig 6 can be withdrawn and we will mention mini CTD results in the text.

13.) Figure 8: Is this the ship CTD data? Why then is not the warmest water on Station 10 and 12 about +3 deg C visible? And

Since we are interested in the water body with oxygen concentration close to saturation, we excluded all TS data points with oxygen well below saturation.

14.) Page 12 – line 10. Please use one temperature throughout a paper. It is fine to use the new conservative temperature, but then you should use it throughout.

Here we use data from a satellite imagery product given in deg C. We will it either mention in the text or convert the data into conservative temperature by assuming a typical salinity for this region.

15.) Page 13 – line 1: "we do not have information on surface salinity or currents." This is exactly the main problem. Very little data is available, and then one cannot really conclude on the suggested processes either. A numerical model could have amended this in a nice way.

We intend to use model simulations in the revised version.

16.) Page 13 – line 6: Also here; "there are some indications that surface water from the Bothnian Sea have been mixed with Bothnian Bay water forming the observed bottom water at station 9". Some anecdotal indications are not really enough to claim that one has new findings worth publishing in an international journal.

See above.

17.) Page 14, line 11-16: While I am no expert in biological processes it is clearly possible that there is growth of organic mater in sea ice, and this should be discussed. A fairly new paper (Assmy et al 2017) also finds that phytoplankton can also grow below

[Figure]

a snow cover.

DCOM in the northern Baltic Sea constitutes of refractory, terrestrially gelbstoff in high concentrations. Autochthonous CDOM plays a minor role only. Therefore, CDOM can be used as a conservative tracer in the northern Baltic Sea.

18.) Page 14, line 18: polyniyas is spelled wrongly.

Will be corrected.

19.) Page 14, line 17 – Page 15, line 7. While this is possible in the Bothnian Bay – you do not have any observations that indicate that this is going on. IF you added some simulations that this is likely, then this text could remain – otherwise it should be deleted.

We will check when the model simulations are done.

20.) Page 15, line 8 – 20. This section finally contains some calculations about the brine water "hypothesis". The calculations appear OK - but does not use a proper range in forcing and boundary conditions. How representative is the 0.2 m of ice thickness? Is there any freshwater discharge during winter?

In the revised text, forcing and boundary conditions will be checked carefully and justified.

---

## Author Comment (AC3) · 26 Nov 2019

First of all, we would like to thank the anonymous referee for the thorough review of our manuscript.

The common concern of all referees is that the amount of data collected during a cruise in the sea ice covered northern Baltic Sea is not sufficient to justify any of the hypotheses raised in the manuscript. Indeed, only three stations could be explored and we hardly can increase the number of in-situ observations. However, the referees proposed an option to save the manuscript by a complementary modeling approach.

We decided to follow this line and will perform a model simulation for the winter season 2016/2017. We will set up a model for the Baltic Sea with earmarked water masses allowing us to identify the origin of water which eventually arrives in the deep water of the Bothnian Bay. Nevertheless, we want to stress that recent ocean models are

not able to reproduce a possible haline convection due to brine release. However, this approach will prolong a revised version of our manuscript and will include substantial changes.

In the following, we respond to the referee's specific remarks. Remarks are shown and followed by our response.

Review #3:

1.) This paper tries to justify that oxygen rich bottom water found in one profile collected in the Bothnian Bay, may have been formed by inflowing water from the Bothnian Sea mixed with surface water in the Northern Quark, and not by salt release from sea ice. The data set is very small, and the processing done poorly explained. It is also hard from the discussion to grasp that the above explanation is what the authors want to say. To make this manuscript more readable they should state more clearly in the discussion whether each explanation they try own ends up with a plausible explanation. I do not suggest any places where to do this in particular, but both the abstract and discussion and conclusions should become clearer. Perhaps add your hypotheses at the end of your introduction. Than it would be easier to state of your data support or do not support each of the hypotheses.

We thank the referee for the useful suggestions. In a revised manuscript, we will include model simulations and formulate hypotheses and there acceptation/rejection more clearly.

2.) Page 3, line 12: Explain how you collected the brine.

We will do it in the revised manuscript.

3.) Page 4, Figure 3 caption: at (red crosses) to stations 7-10.

We will do it in the revised manuscript.

4.) Page 4, lines 1-2: Explain more elaborate how you measured salinity in both ice
core samples (Table 1) and brine samples (Table 2). For instance how big did the samples have to be to measure salinity with the CTD. Also, with the low number of samples you collected, why not measure all with the Guildline?

We have analyzed one ice core from each station and the brine samples with the Guildline and all samples with the mini CTD. A detailed description of the procedure will be given in the revised manuscript.

5.) Page 4, lines 2-7: The Guildline Autosal is a standard instrument used with a standard procedure, so this procedure does not need elaborate description. How you collected ice samples and brine and measured their salinity is on the other hand not standard procedure and needs better description.

See above comment.

6.) Page 5, line 10: Justify how you can assume that 'upper 5m are well mixed and saturated with oxygen'. This might be ok for late winter, although I do not have any reference to recommend.

The only justification is that density and oxygen profiles are homogeneous in the 5m surface layer and do not show any stratification. The assumption is that in winter oxygen production/consumption is negligible compared to surface flux.

7.) Page 6, lines 13-14: Can you justify the assumption that all the rejected salt is trapped in brine pockets inside the ice? Some of it can be released into the water column.

Most of salt is released into water already. (2.4g of 3.0g salt in 1kg frozen sea water) We will make it more clearly in the revised text.

8.) Page 6, line 14: You can hardly regard an average of two brine samples an average (14.7g/kg). At least, remove the decimal.

The referee is right; we will present the result more conservative.

9.) Page 7, line 2: from where and to where is the water 'out-flowing'. I would find the term 'in-flowing' more appropriate if it flows from the Bothnian Sea into the Bothnian Bay.

Here we mean low saline water from the Bothnian Bay layers above Bothnian Sea water. In the revised manuscript, we will describe it precisely.

10.) Page 7, line 2, last words: Change to 'A weak stratification'

Will be done in the revised manuscript.

11.) Page 9 and 10, Figures 6 and 7: Try to use colors that are more easy to separate from each other. Especially Stations 9 and 10 with purple colors.

We will improve the figures clarity.

12.) Page 9, line 6: Mixing lines do not 'show' water masses. They indicate along which line a mix between two source water types can be placed.

We will improve our sloppy formulation.

13.) Page 10, Figure 8: It is confusing when the end points of the mixing lines go beyond the source water masses. It is clear where the end water mass is, but not the source water masses. You should also indicate better which 'greenish dashed and dash-dotted' lines you are referring to in each case. Where is the brine in this figure, having which temperature? Etc.

We will rework the figure and probably split into two.

14.) Page 11, Figure 9: in this figure, the colors of stations 10 and 12 are difficult to distinguish.

The referee probably means stations 7 and 10. As for Fig 8, we will improve the clarity.

15.) Page 11, line 3: The brine must be way beyond the axis in Figure 8. Again, which TS characteristics do you assume in the brine?

Yes, brine is outside the figure. We used our observations as TS characteristics and will give the numbers explicitly in the description of the TS diagram in the revised manuscript.

---

## Author Response (AR1)

IOW, Seestraße 15, 18119 Rostock

Dr. Thomas Neumann

Senior Scientist

Seestraße 15
D-18119 Rostock
phone: +49 381 51 97 130
fax: +49 381 51 97 440
www.io-warnemuende.de
thomas.neumann@
io-warnemuende.de

Rostock, 22.04.2020

**Revision os-2019-48**

Dear John Huthnance (OS editor)

Hereby, I submit the revised version of the manuscript os-2019-48

**Ventilation of the Northern Baltic Sea**

by Thomas Neumann, Herbert Siegel, Matthias Moros, Monika Gerth, Madline Kniebusch, and Daniel Neumann.

First of all, we would like to thank the three anonymous referees and you for the careful consideration of the manuscript and many helpful comments. Following the referees' suggestions, we have extended the study with a numerical model experiment. We have taken into account every remark and worked through the whole text and think, the manuscript has much improved now.

An appendix with our detailed response to the reviewers' comments is attached at the end of the letter. Thank you for time and effort to consider the submitted manuscript.

With best regards,

Thomas Neumann

Appendix

**Major changes of the revised manuscript:**

The referees' main concern of the manuscript was the low number of observational data supporting our findings. For the revised version, we designed and performed numerical model simulations to increase the available data. The model simulation reproduces the campaign and in addition to temperature and salinity variables, a number of passive tracers, earmarking different water masses, were activated. Model results clearly support our hypothesis that dense bottom water is formed due to mixing of different surface water masses and that brine is not involved in this process. Since these results are well justified by the model simulation, we no longer consider satellite data and *in situ* monitoring data.

We included an assessment for model performance in the northern Baltic Sea as an appendix.

**Detailed response (in blue color) to the comments of the referees**

Referee #1:

Abstract. The abstract is not sufficiently informative. Basically, there is just one sentence, which describes, what this paper is about. I would suggest to exclude all general wording about the Baltic Sea, but to add more about the essence of this paper.

We revised the abstract and more information, especially about the model simulation, is given. General information about the Baltic are reduced.

Page 2, lines 2-3: I would suggest to add more detailed explanation (not just reference on Peterson, 2018), how melting sea ice can produce considerable salt fluxes into the ocean? This is quite a new knowledge, and it is important to explain it in more details.

We added ideas of Peterson (2018) on brine release during melting. Basically, increasing permeability enhances gravity forced drainage.

Page 2, lines 3-4: The references to Aagaard,et al., (1981) and Skogseth et al. (2008) are not relevant in this context. These papers describe specific events of dense water formation and do not consider general theory of this event at all. Taking into account

that the authors are considering horizontal advection, as the most probable mechanism of the surface water densification in the Bothnian Bay, I would suggest them to read papers by Shapiro et al (2003) and Ivanov et al. (2004), which summarized all known mechanisms of dense water formation and cascading (not only in the Arctic, but worldwide).

We would like to thank the referee for additional literature recommendations. Now, the introduction starts with the general consideration of deep water ventilation. However, we think that examples of case studies from sea ice regions are useful for our study.

Page 15, lines 19-20: "In areas shallower than the pycnocline, dense water can accumulate at the sea floor and form density driven plumes guided by topography" This is very speculative statement, which is not confirmed by the provided measurements, but only by the references to the older studies, where this idea was also rather claimed, but not strongly supported.

The referee is right. We skipped this statement, since the model simulation clearly excludes such mechanism for ventilation.

Referee #2:

General comments:

- The layer close to bottom at one station (station 9) is marginally interesting. Clearly one would have wanted more than one station to show the persistence of any suggested process.

This argument is certainly true but there are not so many observations. The model simulation, now available, gives more confidence in the findings and show that oxygenated bottom water layer at one station was just the beginning of the winterly ventilation process.

This water is high in oxygen, but also 0.5 deg C above freezing. This indicates that it is not related to sea ice formation, or brine driven convection.

Here we disagree with the referee. When brine mixes with surface water and vertically subsides, it preserves most probably the surface temperature. If the new water mass

then horizontally spreads or moves downslope, entrainment of warmer deep water changes the temperature. However, we do not discuss this process anymore since it was rejected by the model simulation.

So why is all the sea ice observations included? Really – it has no use as the paper is written at the moment. Obviously the authors would have liked to find evidence for the "brine hypothesis" – but they have not.

Yes, this was one of the motivations for the sea ice observations. We wanted to test the hypothesis whether brine contributes to deep water formation in the northern Baltic. We have formulated the hypotheses more pronounced in the revised manuscript.

Additionally, only little data are available for this region from wintertime. Thus, our intention was also to make this observations public available.

- There is generally a small number of citations given. While it is good practice not to overflow with too many, here it is on the sparse side. And one suspects that the authors have spent a limited effort on finding relevant studies. A good example is for the experimental studies in polynyas (Page 2, line 5). Clearly there are many more observations available from polynyas, both in the Arctic and Antarctica. As noted by the other reviewer are also some literature on the down-flow required. Examples on earlier polynya studies are given at the end. In general is there also much more available studies of Baltic sea ice available, where the few seas ice samples could be compared to.

We thank the referee for this recommendation. More general literature on down-flow processes and Baltic specific literature are included in the introduction.

Specific Comments:

Page 2, line 7. The Arcic is a name, should always be spelled with capital A. Correct throughout.

Changed accordingly.

Page 3, line 6: Use of "Fast ice" is wrong. Fast ice means sea ice frozen onto the shore. Here I think you mean pack ice?

https://www.jcomm.info/index.php?option=com_oe&task=viewDocumentRecord&docID=14598

We agree, "pack ice" is right. Changed accordingly.

Page 3, line 11 It is not clear how you sample the brine. You state that:" holes approximately half the depth of the ice thickness were drilled to collect brine". Do you mean that you take out the core, and wait for water to drain back into the whole? How do you know this is the brine? The brine salinity is very tricky to sample, and conditions here are very special with the super low surface salinities.

The brine sampling was like the referee assume. We have described the sampling procedure more explicitly. We are quite sure that we sampled brine. What else could have caused the high, measured salinity? Of course, the sample could be polluted by snow. However, we took care and remove snow from the vicinity of the sampling station.

Page 4 – lower 4 lines. You simply state you used the (standard) Guildline's Autosal 8400B and the accuracy. It is a standard procedure in the field.

Changed accordingly.

Page 4 –line 8: Are you sure you closed the bottles on the way down? With higher pressures this would lead to the bottles imploding, so the standard is to do this on the way up.

Yes, we are sure; exception is the surface bottle because of possible air bubbles included. Doing so, we prevent samples from the wake of the down cast. Pressure differences are not so large in the shallow Baltic Sea.

Page 6 –line 7: What do you mean by; "CTD and salinometer measurements of the melted ice core water are very close and, therefore, the CTD measurements appear to be reliable"? How can you take a CTD measurement of the melted ice core? A CTD needs to be fully submerged in ocean water to work, and measures the conductivity over a much larger volume of water that is inside the conductivity cell. . .

We used the handheld (mini) CTD for these measurements. The conductivity cell is very small compared to the sample water volume. We described the procedure more explicitly in the "methods" section.

Page 6 – line 12; "The mean sea ice bulk salinity in the Bothnian Sea is about 0.6g kg-1". This is a very strong claim when you have ice cores from 3 locations. . ...

We reformulated this statement: - mean sea ice bulk salinity of the sampled cores -

Page 8, Figure 4 caption: ice sheets – this means the large piece of ice on Greenland and Antarctica. You may mean "ice core"?

We call it "ice core slices" now to prevent misunderstanding.

Page 10. Figure 7. The mini CTD observations appear close to the ship-born CTD. If they are plotted in the same figure – then one could see if there are any differences – but this appears not to be the case. This figure is not valuable – unless there are some significant differences – and then these should be shown in Figure 6.

We thank the referee for this recommendation. Indeed, differences are not visible in a figure. Therefore, we skipped a figure with data from the mini CTD and mentioned that the results are very close instead.

Figure 8: Is this the ship CTD data? Why then is not the warmest water on Station 10 and 12 about +3 deg C visible? And

Since we are interested in the water body with oxygen concentration close to saturation, we excluded all TS data points with oxygen well below saturation. Therefore, these data are not shown in Fig. 8. It is explained in the text.

Page 12 – line 10. Please use one temperature throughout a paper. It is fine to use the new conservative temperature, but then you should use it throughout.

Changed accordingly.

Page 13 – line 1: "we do not have information on surface salinity or currents." This is exactly the main problem. Very little data is available, and then one cannot really conclude on the suggested processes either. A numerical model could have amended this in a nice way.

We are very grateful for suggesting a complementary model study. Model results make the study much more concise now.

Page 13 – line 6: Also here; "there are some indications that surface water from the Bothnian Sea have been mixed with Bothnian Bay water forming the observed bottom water at station 9". Some anecdotal indications are not really enough to claim that one has new findings worth publishing in an international journal.

We think, with the help of numerical simulations, the status of "anecdotal indications" has been left.

Page 14, line 11-16: While I am no expert in biological processes it is clearly possible that there is growth of organic mater in sea ice, and this should be discussed. A fairly new paper (Assmy et al 2017) also finds that phytoplankton can also grow below a snow cover.

The referee is right; phytoplankton can grow within and below sea ice. However, the vast majority of CDOM in the northern Baltic Sea constitutes of refractory, terrestrially gelbstoff in high concentrations. Autochthonous CDOM plays a minor role only. Therefore, CDOM can be used as a conservative tracer in the northern Baltic Sea (e.g., Harvey, 2015).

Page 14, line 18: polyniyas is spelled wrongly.

Changed accordingly.

Page 14, line 17 – Page 15, line 7. While this is possible in the Bothnian Bay – you do not have any observations that indicate that this is going on. IF you added some simulations that this is likely, then this text could remain – otherwise it should be deleted.

The model simulation show that brine is not contributing to deep water ventilation. We deleted this text.

Page 15, line 8 – 20. This section finally contains some calculations about the brine water "hypothesis". The calculations appear OK - but does not use a proper range in forcing and boundary conditions. How representative is the 0.2 m of ice thickness? Is there any freshwater discharge during winter?

Since the brine hypothesis has been rejected, we deleted these calculations.

Suggested citations – there are many more:

We are very grateful for the comprehensive list of recommended studies which we have considered carefully.

Referee #3

This paper tries to justify that oxygen rich bottom water found in one profile collected in the Bothnian Bay, may have been formed by inflowing water from the Bothnian Sea mixed with surface water in the Northern Quark, and not by salt release from sea ice. The data set is very small, and the processing done poorly explained. It is also hard from the discussion to grasp that the above explanation is what the authors want to say. To make this manuscript more readable they should state more clearly in the discussion whether each explanation they try own ends up with a plausible explanation. I do not suggest any places where to do this in particular, but both the abstract and discussion and conclusions should become clearer. Perhaps add your hypotheses at the end of your introduction. Than it would be easier to state of your data support or do not support each of the hypotheses.

We thank the referee for the careful consideration of our manuscript and the useful suggestions. In the revised manuscript, we included model simulations and formulate hypotheses and there acceptation/rejection more clearly.

Page 3, line 12: Explain how you collected the brine.

Changed accordingly.

Page 4, Figure 3 caption: at (red crosses) to stations 7-10.

Changed accordingly.

Page 4, lines 1-2: Explain more elaborate how you measured salinity in both ice core samples (Table 1) and brine samples (Table 2). For instance how big did the samples have to be to measure salinity with the CTD. Also, with the low number of samples you collected, why not measure all with the Guildline?

We explained the salinity measurements more detailed.  E.g., 100ml samples were sufficient for salinity measurements with the mini CTD. First, we did test measurements with both Autosal and mini CTD. Since both methods delivered the same data, we decided not to measure all samples with the Autosal.

Page 4, lines 2-7: The Guildline Autosal is a standard instrument used with a standard procedure, so this procedure does not need elaborate description. How you collected ice samples and brine and measured their salinity is on the other hand not standard procedure and needs better description.

Changed accordingly.

Page 5, line 10: Justify how you can assume that 'upper 5m are well mixed and saturated with oxygen'. This might be ok for late winter, although I do not have any reference to recommend.

The only justification is that density and oxygen profiles are homogeneous in the 5m surface layer and do not show any stratification.  The assumption is that in winter oxygen production/consumption is negligible compared to surface flux. We explicitly mention the "no stratification" criterion in the text.

Page 6, lines 13-14: Can you justify the assumption that all the rejected salt is trapped in brine pockets inside the ice? Some of it can be released into the water column.

Most of salt is released into water already. (2.4g of 3.0g salt in 1kg frozen sea water is released to the water column) We made this assumption more clearly in the revised text.

Page 6, line 14: You can hardly regard an average of two brine samples an average (14.7g/kg). At least, remove the decimal.

We agree. It is the mean of **our** samples and we removed the decimal.

Page 7, line 2: from where and to where is the water 'out-flowing'. I would find the term 'in-flowing' more appropriate if it flows from the Bothnian Sea into the Bothnian Bay.

Here we mean low saline water from the Bothnian Bay layers above Bothnian Sea water. We improved the wording.

Page 7, line 2, last words: Change to 'A weak stratification'

Changed accordingly.

Page 9 and 10, Figures 6 and 7: Try to use colors that are more easy to separate from each other. Especially Stations 9 and 10 with purple colors.

We use now green for station 9 und blue for station 10.

Page 9, line 6: Mixing lines do not 'show' water masses. They indicate along which line a mix between two source water types can be placed.

We thank the referee for this hint and improved our sloppy formulation.

Page 10, Figure 8: It is confusing when the end points of the mixing lines go beyond the source water masses. It is clear where the end water mass is, but not the source water masses. You should also indicate better which 'greenish dashed and dash-dotted' lines you are referring to in each case. Where is the brine in this figure, having which temperature? Etc.

We revised the figure. Brine is outside the figure. The used TS properties of brine are given in the figure caption and text.

Page 11, Figure 9: in this figure, the colors of stations 10 and 12 are difficult to distinguish.

The referee probably means stations 7 and 10. We note in the figure caption that station 7 data are mostly covered by station 9 and 10 data. Station 10 is in blue color and station 12 in violet color.

Page 11, line 3: The brine must be way beyond the axis in Figure 8. Again, which TS characteristics do you assume in the brine?

The assumed TS characteristics are now given in the text and figure caption.

Page 11, line 15: 'mechanisms'

Changed accordingly.

Page 13, line 5: 'weakening'

Changed accordingly.

Page 14, line 18: 'polynyas'

Changed accordingly.

Page 15: make it clearer which formation process you trust and which you do not trust.

Changed accordingly.

[revised manuscript text omitted]

---

## Author Response (AR2)

IOW, Seestraße 15, 18119 Rostock

Dr. Thomas Neumann

Senior Scientist

Rostock, 14.05.2020

Seestraße 15 D-18119 Rostock phone: +49 381 51 97 130 fax: +49 381 51 97 440 www.io-warnemuende.de thomas.neumann@ io-warnemuende.de

Revision os-2019-48

Dear John Huthnance (OS editor)

Hereby, I submit the revised version of the manuscript os-2019-48

**Ventilation of the Northern Baltic Sea**

by Thomas Neumann, Herbert Siegel, Matthias Moros, Monika Gerth, Madline Kniebusch, and Daniel Neumann.

First of all, we would like to thank the two anonymous referees and you for the careful consideration of the manuscript and many helpful comments. We followed the referees' suggestions and editorial comments.

An appendix with our detailed response to the referees' comments is attached at the end of the letter. Thank you for time and effort to consider the submitted manuscript.

With best regards,

Thomas Neumann

**Appendix**

Response (in blue color) to the comments of the referees:

I had a reviewer-type question of my own: whether the distinct signal at station 9 (only) on March 12-13 is consistent with the progress of the modelled surface tracer. This is not very clear in figure 10 where stations 7, 9, and 10 all seem to be on the edge of the tracer signal.

This is true; the model shows the signal simultaneously at stations 9 and 10 while at station 7 the signal is delayed by two weeks. Observations do not show the signal at station 10. One can now speculate that CTD data from station 10 show the very beginning of the signal: Decreasing temperature and increasing oxygen in the bottom layer which is not seen for station 7 (Fig. 7). We think the evidence is too weak for a discussion in this direction.

Furthermore, model results are associated with an uncertainty. We cannot be sure that the model is completely right. E.g., the model speed of the plume may be different from the real speed resulting in a different timing.

**Referee #2**

I suggest your re-write the abstract, just to summarize it all in a better way. The name of the ship is not essential information and does not belong in the abstract, but the name of the model should be there. The abstract should describe your findings. Here it is really only the 3 last lines that describe the findings."

We have separated the abstract into 3 parts: (i) introduction into the problem, (ii) collected data and used methods, and (iii) main results. We do not mention the name of the vessel anymore.

We believe that the name of the model is not essential for the abstract. Many models could have been used for this investigation. This is a technical aspect described in the methods section.

Referee #3 A few comments follows: Page 3 Line 10: CDT should be CTD Changed accordingly.

Page 4 Figure text to Figure 4: ... a CTD cast 'was' performed (not 'were') Changed accordingly.

Page 6 Line 32: is 'fraction' a better word than 'share'? Changed accordingly.

Page 14 Should paragraph lines 12-13 perhaps be before paragraph lines 9-11?" We changed the order.

My detailed Editor comments Page 1 Abstract last sentence. Better "Brine barely contributes to dense bottom water." Line 20. (Raateoja, 2013) is not now very "recent". This is true – changed into "last decades"

Line 25. "a remarkable" -> "a significant" or "much" Changed accordingly.

Lines 3-4. "on the way of the dense water plume" -> "on its way by the dense water plume." or "en route by the dense water plume." Changed accordingly.

Lines 15-16. "re-suspension" (spelling) Changed accordingly.

Line 17. "dense" (spelling) Changed accordingly.

Line 26. "allow for a" -> "allows" Changed accordingly.

Line 32. ". . consist . ." Changed accordingly.

Page 5 Line 4 Maybe ". . brine draining from the ice accumulated in the hole . ."? Changed accordingly.

Page 11 Line 17. Maybe ". . solid lines indicate mixing of water masses . ." Changed accordingly. Line 22 and figure 8 caption. -0.78 or -0.79? -0.79 is correct

Line 22. ".. constraints . ." (spelling) Changed accordingly.

Lines 26-27. Please relate "oxygen saturated" to the rest of the sentence. We rephrased the sentence.

Page 12 Line 3. "drops down" -> "decreases" Changed accordingly.

Line 5. "ago" -> "before" Changed accordingly.

Line 6. "leaving the surface water" -> "from the surface"? Changed accordingly.

Line 11. ". . sea-ice-covered . ." Changed accordingly.

Page 13, figure 10 caption. ". . next-to-bottom . .". This is not quite clear. Do you mean the bottom model layer or the next one up? bottom model layer, changed accordingly.

Page 14 line 13. "descent" -> "descend" Changed accordingly.

Page 16 Line 5. ". . initiates . ." Changed accordingly.

Line 6. ". . shelves . ." Changed accordingly.

Line 11. ".. Northern ..." Delete "a" Changed accordingly.

Line 16. "assume" -> "find"? Changed accordingly.

 Line 25. "establishes" -> "is established" Changed accordingly.

Page 17 Figure A1 caption. "extent" (spelling) Changed accordingly.

Page 19 lines 1 and 5. "reasonably" (spelling) Changed accordingly.

**Ventilation of the Northern Baltic Sea**

Thomas Neumann1, Herbert Siegel1, Matthias Moros1, Monika Gerth1, Madline Kniebusch1, and Daniel Neumann1

[revised manuscript text omitted]